# Organ-Specific and Mixed Responses to Pembrolizumab in Patients with Unresectable or Metastatic Urothelial Carcinoma: A Multicenter Retrospective Study

**DOI:** 10.3390/cancers14071735

**Published:** 2022-03-29

**Authors:** Takuto Shimizu, Makito Miyake, Nobutaka Nishimura, Kuniaki Inoue, Koyo Fujii, Yusuke Iemura, Kazuki Ichikawa, Chihiro Omori, Mitsuru Tomizawa, Fumisato Maesaka, Yuki Oda, Tatsuki Miyamoto, Keiichi Sakamoto, Keisuke Kiba, Masahiro Tanaka, Nobuo Oyama, Eijiro Okajima, Ken Fujimoto, Shunta Hori, Yosuke Morizawa, Daisuke Gotoh, Yasushi Nakai, Kazumasa Torimoto, Nobumichi Tanaka, Kiyohide Fujimoto

**Affiliations:** 1Department of Urology, Nara Medical University, Kashihara 634-8522, Japan; t-shimizu@naramed-u.ac.jp (T.S.); k166559@naramed-u.ac.jp (K.I.); k150814@naramed-u.ac.jp (K.S.); horimaus@naramed-u.ac.jp (S.H.); morizawa@naramed-u.ac.jp (Y.M.); dgotou@naramed-u.ac.jp (D.G.); nakaiyasushi@naramed-u.ac.jp (Y.N.); torimoto@naramed-u.ac.jp (K.T.); kiyokun@naramed-u.ac.jp (K.F.); 2Department of Urology, Okanami General Hospital, Iga 518-0842, Japan; ffxxxx.nqou@gmail.com (N.N.); kenfujinara@hotmail.co.jp (K.F.); 3Department of Urology, Osaka Gyoumeikan Hospital, Osaka 554-0012, Japan; mitsu32jp2001@gmail.com; 4Department of Urology, Hirao Hospital, Kashihara 634-0076, Japan; k197447@naramed-u.ac.jp; 5Department of Urology, Takai Hospital, Tenri 632-0372, Japan; k161356@naramed-u.ac.jp; 6Department of Urology, Nara Prefecture General Medical Center, Nara 630-8581, Japan; k110140@naramed-u.ac.jp; 7Department of Urology, Yamato Takada Municipal Hospital, Yamato Takada 635-8501, Japan; k178754@naramed-u.ac.jp; 8Department of Urology, Nara City Hospital, Nara 630-8305, Japan; k192761@naramed-u.ac.jp (F.M.); e-okajima@nara-jadecom.jp (E.O.); 9Department of Urology, Nara Prefecture Seiwa Medical Center, Ikoma 636-0802, Japan; k197610@naramed-u.ac.jp (Y.O.); nobuo.seiwa@nifty.com (N.O.); 10Department of Urology, Hoshigaoka Medical Center, Hirakata 573-8511, Japan; K143110@naramed-u.ac.jp; 11Department of Urology, Kindai University Nara Hospital, Ikoma 630-0293, Japan; 151289@med.kindai.ac.jp; 12Department of Urology, Osaka Kaisei Hospital, Osaka 532-0003, Japan; masa-t@kaisei-hp.co.jp; 13Department of Brachytherapy, Nara Medical University, Kashihara 634-8522, Japan; sendo@naramed-u.ac.jp

**Keywords:** organ-specific response, mixed response, bladder cancer, urothelial carcinoma, pembrolizumab, immunotherapy, immune checkpoint inhibition, anti-PD-1

## Abstract

**Simple Summary:**

To investigate the organ-specific responses and clinical outcomes of mixed responses (MRs) to treatment with pembrolizumab for unresectable or metastatic urothelial carcinoma (ur/mUC), 136 patients were analyzed retrospectively. The total objective response rate (ORR) and organ-specific ORR were determined according to RECIST version 1.1 as follows: (i) CR, (ii) PR, (iii) SD, and (ⅳ) PD. MR was defined as when PD occurred in one lesion plus either CR or PR in other lesion simultaneously, and 12 cases were applicable. Most of the organ-specific ORR was 30–40%, but bone metastasis was only 5%. Compared to non-responders, responders to locally advanced lesions and lymph node, lung, or liver metastases were involved in OS, but local recurrence and bone metastases were not involved in OS. In MR, patients who continued pembrolizumab experienced longer survival times compared to patients who discontinued pembrolizumab and received standard treatment.

**Abstract:**

To investigate the organ-specific response and clinical outcomes of mixed responses (MRs) to immune checkpoint inhibitors (ICIs) for unresectable or metastatic urothelial carcinoma (ur/mUC), we retrospectively analyzed 136 patients who received pembrolizumab. The total objective response rate (ORR) and organ-specific ORR were determined for each lesion according to the Response Evaluation Criteria in Solid Tumors version 1.1 as follows: (i) complete response (CR), (ii) partial response (PR), (iii) stable disease (SD), and (iv) progressive disease (PD). Most of the organ-specific ORR was 30–40%, but bone metastasis was only 5%. There was a significant difference in overall survival (OS) between responders and non-responders with locally advanced lesions and lymph node, lung, or liver metastases (HR 9.02 (3.63–22.4) *p* < 0.0001; HR 3.63 (1.97–6.69), *p* < 0.0001; HR 2.75 (1.35–5.59), *p* = 0.0053; and HR 3.17 (1.00–10.0), *p* = 0.049, respectively). MR was defined as occurring when PD happened in one lesion plus either CR or PR occurred in another lesion simultaneously, and 12 cases were applicable. MR was significantly associated with a poorer prognosis than that of the responder group (CR or PR; HR 0.09 (0.02–0.35), *p* = 0.004). Patients with bone metastases benefitted less. Care may be needed to treat patients with MR as well as patients with pure PD. Further studies should be conducted in the future.

## 1. Introduction

With the advent of immune checkpoint inhibitors (ICIs), significant strides have been made in the treatment of various malignancies since the era of conventional cytotoxic chemotherapy, helping to improve patient survival. For patients with unresectable or metastatic urothelial carcinoma (ur/mUC), pembrolizumab, which is a programmed death-1 (PD-1) inhibitor, and avelumab, a programmed death ligand-1 (PD-L1) inhibitor, are currently approved and widely used, following the KEYNOTE-045 and JAVELIN Bladder 100 trials [1,2]. The appearance of ICIs has certainly improved the prognosis of ur/mUC; however, the response rate is insufficient [3]. Furthermore, the activity of ICIs may be affected depending on the different immunities of metastatic locations [4,5], and some small and limited reports have shown that the effect of ICIs differs depending on the metastatic organ [6,7]. These factors may complicate the determination of treatment efficacy and strategy. In many cases, similar therapeutic effects are seen in any metastatic lesion; however, some patients show a mixed response (MR) that regresses in some tumors and progresses in other tumors at the same time. This outcome is judged to be progressive disease (PD) according to the Response Evaluation Criteria in Solid Tumors (RECIST) or the immune-related RECIST (irRECIST), which are widely used to determine the effectiveness of treatments [8,9]. However, knowledge of real-world clinical outcomes of MR and the organ-specific objective response rate (ORR) to pembrolizumab in ur/mUC patients is lacking [6,10]. Here, we performed a retrospective, multicenter, collaborative study to compare the outcomes of response patterns in patients treated with pembrolizumab. 

## 2. Materials and Methods

### 2.1. Ethical Approval

This study was approved by the Institutional Review Board of Nara Medical University (Nara, Japan; study protocol ID: NMU-2891) and complied with the 1964 Helsinki Declaration and its later amendments. As the data for the study were obtained through a retrospective review, a waiver of informed consent was approved by the IRB. The personal information of the subjects was anonymized when necessary, and the information was labeled with an identifying code to make it possible to distinguish between individuals. The deidentified patient data were analyzed.

### 2.2. Patient Selection and Data Collection

Between December 2017 and June 2021, 151 patients received pembrolizumab for chemotherapy-resistant ur/mUC in NURTG-affiliated hospitals. A total of 15 patients were excluded because of insufficient radiographic access, leaving 136 patients for inclusion in our retrospective analysis. Clinicopathological variables (age, sex, Eastern Cooperative Oncology Group–performance status (ECOG–PS), primary site, variant histology, number of prior chemotherapies, interval since last chemotherapy, and follow-up period) and metastatic or unresectable lesions, including the primary local lesion at the initial administration of pembrolizumab, were extracted from medical records, computed tomography (CT) scans, or magnetic resonance imaging (MRI) for analysis.

### 2.3. Response Evaluation

The treatment response was assessed using CT or MRI, which was performed every 3–4 cycles of pembrolizumab. If clinically necessary, other radiographic evaluations were performed as appropriate. 

All metastases that measured ≥10 mm in the long axis (lymph node (LN) metastases ≥15 mm in the short axis) on imaging were defined as measurable lesions in accordance with RECIST version 1.1 and assessed before and during pembrolizumab treatment. The response of each organ was measured and evaluated by defining the sum of the long axis of all non-LN metastases or the short axis of all LN metastases as the tumor burden.

For each metastatic organ, the best response was classified as: (i) complete response (CR), disappearance or reduction to <10 mm in the short axis for all LN metastases; (ii) partial response (PR), >30% reduction from baseline; (iii) stable disease (SD), neither CR, PR, nor PD; and (iv) PD, >20% growth from the minimum sum of diameters during the course and >5 mm growth. “Mixed response” was defined as the finding of simultaneous regressing and progressing metastatic lesions (e.g., PD in one lesion plus either CR or PR in another lesion) [10,11].

### 2.4. Statistical Analysis

Statistical analyses were performed using GraphPad Prism 7.0 (GraphPad Software, San Diego, CA, USA). Multiple comparison tests of clinicopathological characteristics were performed using one-way analysis of variance and Kruskal–Wallis tests, as appropriate. Overall survival (OS) curves were estimated using the Kaplan–Meier method. OS was calculated from the day when pembrolizumab was started or when MR was confirmed until the date of the last follow-up or death by any cause. The differences between each group were compared using the log-rank test. Multivariate analyses were performed via logistic and Cox regression analyses using IBM SPSS software version 21 (SPSS Inc., Chicago, IL, USA). Two-sided tests were used in all tests, and a *p*-value < 0.05 was considered statistically significant in all analyses.

## 3. Results

Figure 1 shows the flowchart of the study. We investigated organ-specific ORR in 136 patients with ur/mUC treated with pembrolizumab. In addition, the total ORR was CR in 10 (7.4%), PR in 34 (25.0%), SD in 20 (14.7%), PD in 60 (44.1%), and MR in 12 (8.8%) patients. Table 1 lists the clinicopathological information of 136 patients and 3 groups (responder group, MR, and non-responder group). Multiple comparisons in the three groups showed that there were no statistically significant differences in characteristics other than the observation period.

### 3.1. Organ-Specific ORR Results

Figure 2 shows the organ-specific ORR and Kaplan–Meier curves comparing responders and non-responders for each site of metastasis.

Six unresectable and metastatic sites of local recurrence; local advance; and lymph node, lung, liver, and bone metastases are shown. Because of the small number of metastatic cases, metastatic sites in the brain, pleura, peritoneum, and other locations were excluded. For lesion sites other than bones, the organ-specific ORR was 30–40%, but the ORR for bone metastasis was only 5%. There was no significant difference in OS between responders and non-responders with local recurrence and bone metastasis metastases (hazard ratio (HR) 2.86 (0.67–12.3), *p* = 0.15; HR 3.44 (0.38–31.1), *p* = 0.27, respectively). However, there was a significant difference in OS between responders and non-responders with locally advanced lesions and lymph node, lung, or liver metastases (HR 9.02 (3.63–22.4), *p* < 0.0001; HR 3.63 (1.97–6.69), *p* < 0.0001; HR 2.75 (1.35–5.59), *p* = 0.0053; HR 3.17, (1.00–10.0), *p* = 0.049, respectively). Figure 3 shows the change in organ-specific ORR over time. A relatively long-term response was observed in local lesions and lymph node metastases, but a short-term response was observed in lung and liver metastases. In other words, lung and liver metastases tend to become refractory, even if an effect is initially observed. Appendix A shows the univariate and multivariate analyses of background factors for OS. Univariate analysis revealed that ECOG–PS ≥ 2, liver metastasis, and bone metastasis were statistically significant poor prognostic factors for OS (HR 5.53 (2.20–13.9), *p* = 0.0003; HR 2.16 (1.08–4.31), *p* = 0.030; HR 2.36 (1.23–4.55), *p* = 0.010, respectively). Multivariate analysis revealed that ECOG–PS ≥ 2 was an independent prognostic factor for OS (HR 2.29 (1.20–4.36), *p* = 0.012). 

### 3.2. Results of Research on MR

A total of 31 (22.8%) of the 136 patients who received pembrolizumab had different responses in different target organs, as follows: CR + PR in 3 (2.2%), CR + SD in 3 (2.2%), PR + SD in 3 (2.2%), SD + PD in 10 (7.4%), PD + CR in 2 (1.5%), PD + PR in 6 (4.4%), PD + SD + PR in 2 (1.5%), and PD + CR + PR in 2 (1.5%). Among these responses, 12 met the MR criteria (Figure 1).

Figure 4 shows representative cases of MR with simultaneous progression and regression following pembrolizumab treatment. In Case A, CR of lung metastasis and PD of local lesions were simultaneously observed 6 months after administration of pembrolizumab. In Case B, CR of lung and liver metastases and PD of rectal metastases were simultaneously observed 4 months after administration of pembrolizumab.

Survival curves demonstrated that the curve of the MR group was similar to that of the responder group showing CR or PR, and the non-responder group showing SD or PD (Figure 5a). There was a significant difference in OS between the responder and MR groups (HR 0.09 (0.02–0.34), *p* = 0.004). There was no significant difference in OS between the MR and non-responder groups (HR 1.69 (0.89–3.23), *p* = 0.11). Figure 5b shows the Kaplan–Meier curve of the five groups (CR, PR, MR, SD, and PD). The MR group had a significant difference in OS compared to the CR and PR groups (HR 0.10 (0.02–0.40), *p* = 0.0012 and HR 0.19 (0.06–0.63), *p* = 0.0068, respectively) and showed a tendency to have a significant difference from that the PD group (HR 1.89 (0.97–3.67), *p* = 0.062). No significant difference was found in SD (HR 1.40 (0.58–3.41), *p* = 0.46). 

Appendix A shows the organ-specific response to pembrolizumab in 12 patients with MR, and swimmer plot images depict the clinical course after pembrolizumab administration. During the follow-up period, 8 (75.0%) of the 12 patients died of urothelial cancer (UC). Figure 6 shows the Kaplan–Meier curves of the 12 patients who continued and discontinued pembrolizumab from the time of MR diagnosis. In the MR group, patients who continued pembrolizumab after the diagnosis of MR had longer survival than those who discontinued pembrolizumab and received standard treatment (best supportive care or chemotherapy) (HR 8.8 (1.28–60.2), *p* = 0.03).

## 4. Discussion

In this multicenter retrospective study, we analyzed the response of patients with ur/mUC to pembrolizumab after the failure of platinum-based chemotherapy. In Japan, avelumab was recently approved as an ICI for UC in addition to pembrolizumab [2], but the cohort in this study did not include cases of avelumab use. In this organ-specific ORR study, the poor response to pembrolizumab of bone metastasis was prominent (Figure 2). Bone metastases in UC have a poorer prognosis than those in other genitourinary cancers (prostate or renal cell cancer), as previously reported by our group [12]. Bone metastasis is an important condition that significantly impairs quality of life (QOL) when skeletal-related events (SREs) occur [13]. Some efficacy of bone-modifying agents, such as zoledronic acid or denosumab, has been reported for SREs, but it is not sufficient [14]. Although the research direction of ICIs for ur/mUC is scattered, there are few reports focusing on bone metastasis, and it may be difficult to evaluate because other therapeutic modalities, such as radiation therapy, may be added [13,15,16]. The advent of pembrolizumab has not led to significant progress in the treatment of bone metastasis (Figure 2), and treatment and maintenance of the QOL of patients with UC-related bone metastases should be investigated further in the future. Local lesions and lymph node metastases tended to have a longer sustained response to pembrolizumab than visceral metastases (Figure 3). Lymph nodes are characterized by the close interaction of immune cells with antigen-presenting cells and representatives of the adaptive immune system, including B and T cells. Recent studies suggest that tumor-infiltrating lymphocytes within lymph node metastases are associated with a better response rate to ICIs [17]. In addition, the liver is thought to be an inhibitory immunomodulatory organ that can weaken immune response and induce immune tolerance [18,19].

In this MR cohort, MR was common (8.8%). This percentage is not significantly different from that reported in other studies. Furubayashi et al. found MR in 4 of 31 cases (12.9%) [10]. Regarding survival curves, the MR group had a curve between that of the PR and SD groups (Figure 5b). Even when comparing responders and non-responders, MR was in the middle (Figure 5a). In addition, the MR curve was located in the middle during the early stage until 12–18 months of follow-up, but then gradually declined and eventually followed the same course as that of the non-responder groups (SD or PD), and it showed a tendency to be different from the curves of responder (CR or PR) (Figure 5a). Finally, there was a statistically significant difference between the MR and responder groups, but there was no significant difference between the MR and non-responder groups. The study of MR following ICI treatment for stage IV melanoma by Rauwerdink et al. also reported that the MR group had a survival curve between that of the responder and non-responder groups [11]. This and previous studies suggest that the MR group may have a slightly longer survival time than the other PD and non-responder groups. Responses to ICIs may also depend on the tumor microenvironment (TME), which is composed of various cell types, such as fibroblasts and immune cells [20]. TME may contribute to the heterogeneity of the response to ICIs, such as differences in organ-specific ORR or individual MR. The immunoinflammatory phenotype, which is essentially characterized by the presence of T cells expressing CD4 and CD8, correlates with a higher response rate to ICIs than that in the immune desert phenotype in which T cells are absent [21]. Sakatani et al. reported that a high infiltration of TMEs by T cells expressing CD8 was significantly associated with a favorable objective response and overall and progression-free survival in patients with ur/mUC treated with pembrolizumab [22]. Furthermore, it has been pointed out that PD-1 inhibitors may also activate regulatory T cells, and the balance of T cells expressing CD8 and regulatory T cells in the TME may regulate the response to PD-1 inhibitors [23,24]. It may affect the activity of ICIs depending on the immunity of different metastatic locations. Chemotherapy before pembrolizumab administration may also be involved in TME. Recent studies have shown that conventional chemotherapy not only has a direct cytotoxic effect on tumor cells, but also promotes an antitumor immune response [25,26,27]. Our group has reported that responsiveness to full-dose gemcitabine plus cisplatin or carboplatin may also correlate with subsequent responsiveness to pembrolizumab [28]. Therefore, regarding the systemic treatment of ur/mUC, a treatment strategy that includes everything from chemotherapy to ICIs will become important for clinicians.

In Japan, enfortumab vedotin was recently approved as a third-line treatment for patients with ur/mUC who failed pembrolizumab [29]. In other words, enfortumab vedotin appears to be the next treatment when PD occurs when using pembrolizumab. However, some PDs, such as the MR examined in this study, may be annoying to clinicians. This is because the clinical outcome of patients with MR may benefit from continued pembrolizumab treatment for a period. In fact, there was a difference in OS between the group that continued pembrolizumab and the group that discontinued pembrolizumab after MR was observed in this study (Figure 6). Nevertheless, it must be considered that there may have been factors, such as the effects of radiation therapy on organs non-responsive to pembrolizumab; however, but it was suggested that multidisciplinary treatment could prolong the prognosis. This could serve as a warning to switch to the next treatment in a timely manner.

This study has some limitations. First, its retrospective nature leads to inherent selection bias. For example, the criteria for determining whether to continue or discontinue pembrolizumab and the timing of treatment changes depended on the institution’s protocol and physician discretion. The cohort was derived from multiple institutions, which can lead to inconsistencies in surgical skill, clinical interpretation, and pathological diagnosis. Second, the small sample size may be one of the reasons why there was no significant difference in OS between the MR and non-responder groups. Third, we did not employ irRECIST or immune-related response criteria (irRC) to evaluate the response of each organ [9,30]. However, neither irRECIST nor irRC discuss MR. In irRECIST, which is more popular than irRC, the basic distinction of CR, PR, SD, and PD conforms with RECIST but takes time to evaluate. In this study, the RECIST evaluation was applied to categorize the reaction of each organ.

## 5. Conclusions

In this multicenter, retrospective study, we confirmed that patients with visceral metastasis had a poor prognosis and the possibility of a short-term response, even if pembrolizumab was effective. Furthermore, MR is common. The MR group had a significantly different OS from that of the CR + PR group; however, there was no statistically significant difference in OS from that of the SD + PD non-response group.

## Figures and Tables

**Figure 1 cancers-14-01735-f001:**
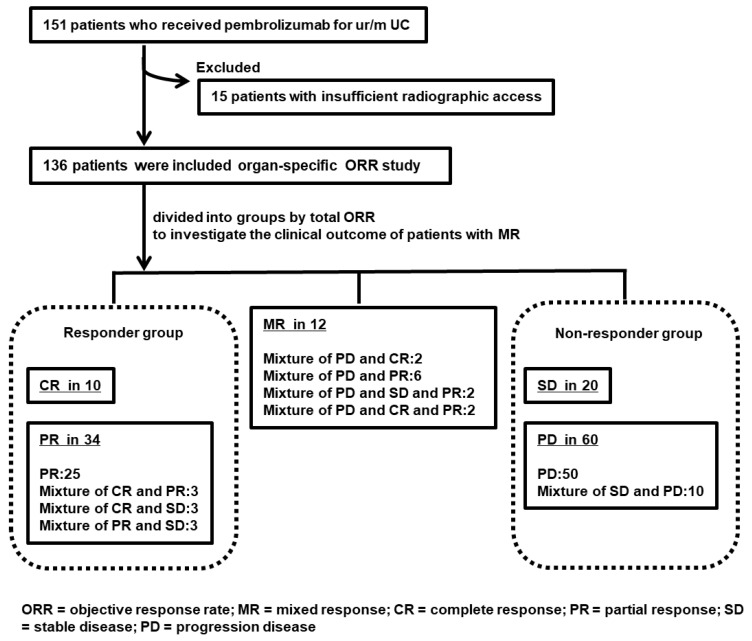
Study flow chart of organ-specific objective response rate (ORR) and mixed response (MR) to pembrolizumab for unresectable or metastatic urothelial carcinoma (ur/mUC). A total of 151 patients with ur/mUC treated with pembrolizumab were enrolled, but 15 were excluded because of a lack of data. The remaining 136 patients were examined for organ-specific ORR and clinical outcome of MR. The responder group included 10 patients with CR and 34 with PR. The non-responder group included 20 patients with SD and 60 patients with PD; there were 12 patients with MR.

**Figure 2 cancers-14-01735-f002:**
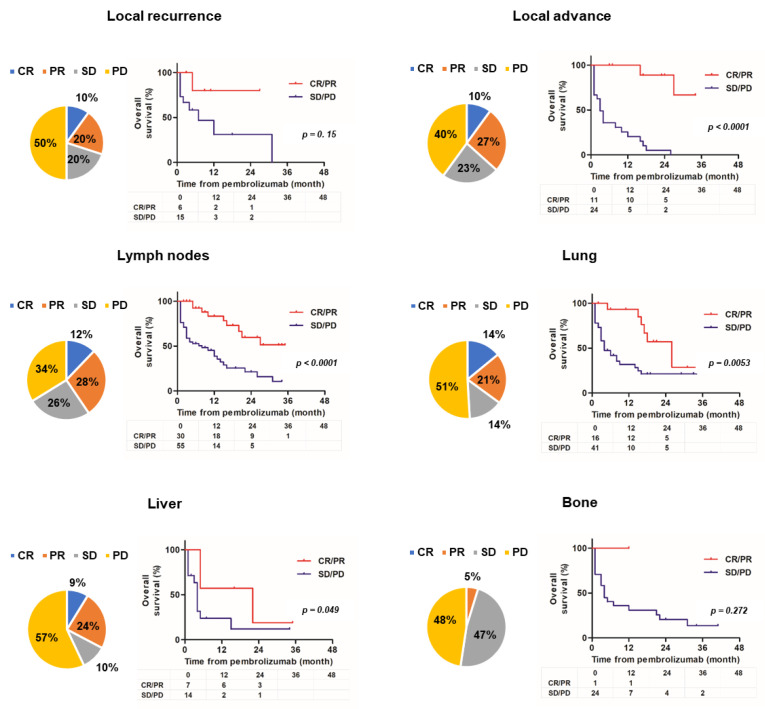
Organ-specific objective response rate and Kaplan–Meier curves comparing responder and non-responder groups. The pie chart presents the best response for each lesion and the Kaplan–Meier estimates of overall survival compared to responders or non-responders for each lesion (local recurrence, local advance, lymph node, lung, liver, and bone).

**Figure 3 cancers-14-01735-f003:**
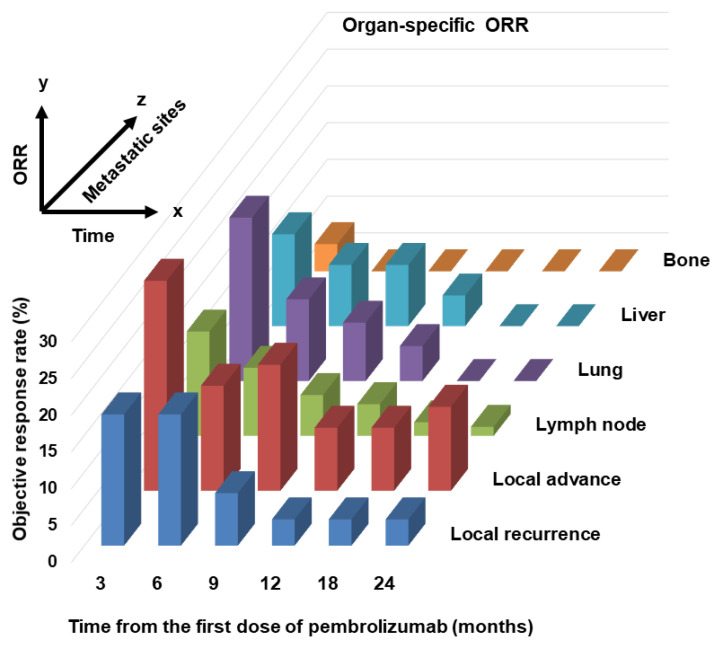
Changes in organ-specific objective response rate (ORR) over time. A three-dimensional image of the passage of time on the *x*-axis, ORR on the *y*-axis, and each lesion (local recurrence, local advance, lymph node, lung, liver, and bone) on the *z*-axis.

**Figure 4 cancers-14-01735-f004:**
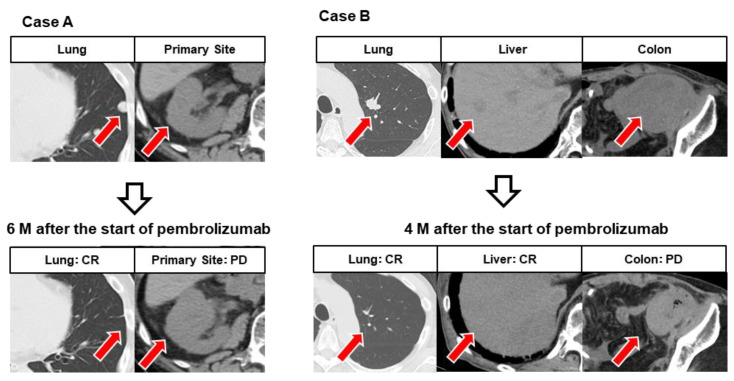
Representative images showing mixed response (MR) in different organs. Case (**A**): After six courses of pembrolizumab, the left lung metastasis had completely disappeared, but the primary lesion of the right renal pelvis showed progression. Case (**B**): After five courses of pembrolizumab, left lung metastases and liver metastases had completely disappeared, while colonic metastases showed progression.

**Figure 5 cancers-14-01735-f005:**
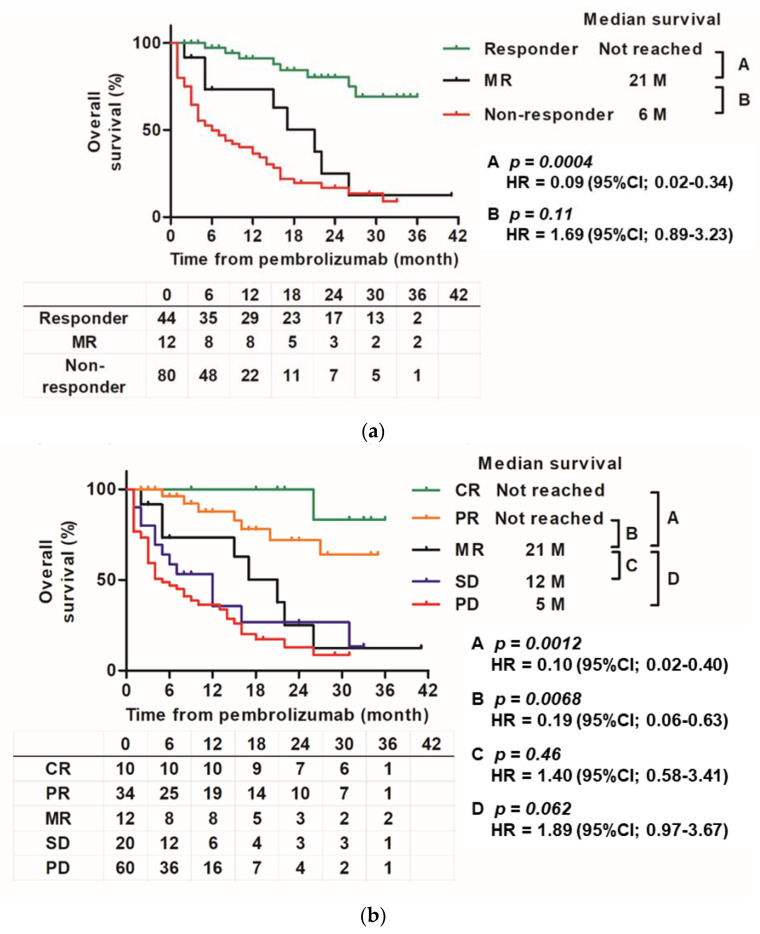
Kaplan–Meier curve for each objective response group. (**a**): Kaplan–Meier estimates of overall survival according to three groups (responder, MR, and non-responder groups) from the start of pembrolizumab treatment to the date of death by any cause. (**b**): Kaplan–Meier estimates of overall survival for five groups (CR, PR, MR, SD, and PD) from the start of pembrolizumab treatment to the date of death by any cause.

**Figure 6 cancers-14-01735-f006:**
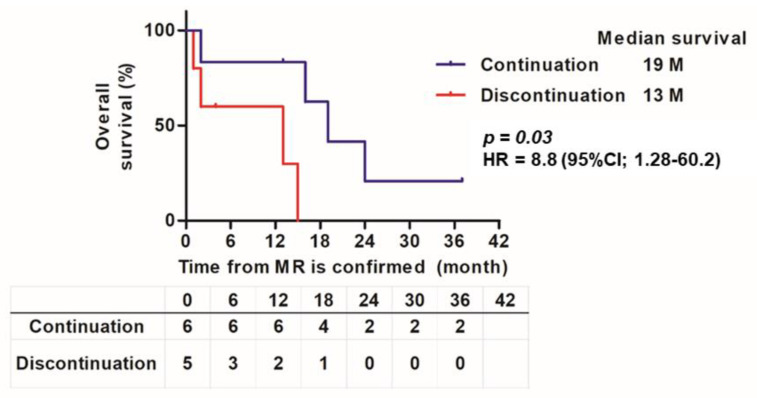
Kaplan–Meier curve of pembrolizumab continuation group and discontinuation group after MR confirmation. Kaplan–Meier estimates of overall survival according to pembrolizumab continuation and discontinuation from when MR was confirmed to the date of death by any cause. In the discontinuation group, pembrolizumab was discontinued and standard treatment (chemotherapy and best supportive care) was provided.

**Table 1 cancers-14-01735-t001:** Clinicopathological background factors of 136 patients and 3 groups.

Variables		Total	Response to Pembrolizumab			*p* Value
			Responder Group CR/PR	MR	Non-Responder Group SD/PD	
Total		136 (100%)	44 (32%)	12 (9%)	80 (59%)	
Age	Median, (range)	73 (49–87)	74 (54–85)	68 (52–82)	72 (49–87)	*0.09*
Sex	Male	100 (74%)	31 (70%)	9 (75%)	60 (75%)	*0.16*
ECOG–PS	0–1 2≤	118 (867%) 18 (13%)	42 (95%) 2 (5%)	10 (83%) 2 (17%)	66 (83%) 14 (17%)	*0.32*
Hemoglobin	≥10 g/dL <10 g/dL	91 (67%) 42 (31%)	28 (64%) 16 (36%)	7 (58%) 5 (42%)	58 (73%) 22 (27%)	*0.15*
eGFR (mL/min/1.73 m^2^)	≥45 <45	73 (54%) 58 (43%)	24 (55%) 20 (45%)	7 (58%) 5 (42%)	46 (58%) 34 (42%)	*0.10*
Primary site	BC UTUC	71 (52%) 65 (48%)	18 (41%) 26 (59%)	8 (67%) 4 (33%)	45 (56%) 35 (44%)	*0.10*
Variant histology	n (%)	13 (10%)	4 (9%)	0 (0%)	9 (11%)	*0.56*
Number of prior chemotherapies	1 2≤	113 (83%) 23 (17%)	34 (77%) 10 (23%)	8 (67%) 4 (33%)	71 (89%) 9 (11%)	*0.18*
Interval since last chemotherapy	<3 months ≥3 months	85 (63%) 51 (37%)	23 (52%) 21 (48%)	8 (67%) 4 (33%)	55 (69%) 25 (31%)	*0.10*
Evaluable lesion						
Local recurrence	n (%)	24 (18%)	8 (18%)	3 (25%)	15 (19%)	*0.28*
Local advance	n (%)	35 (26%)	12 (27%)	1 (8%)	20 (25%)	*0.16*
Lymph nodes	n (%)	85 (63%)	32 (73%)	8 (67%)	46 (58%)	*0.10*
Lung	n (%)	57 (42%)	13 (30%)	9 (75%)	37 (46%)	*0.17*
Liver	n (%)	22 (16%)	5 (11%)	3 (25%)	14 (18%)	*0.28*
Bone	n (%)	25 (18%)	3 (7%)	3 (25%)	18 (23%)	*0.32*
Brain	n (%)	2 (2%)	0 (0%)	2 (17%)	0 (0%)	*0.98*
Pleura	n (%)	5 (4%)	2 (5%)	2 (17%)	2 (3%)	*0.85*
Peritoneal	n (%)	11 (8%)	1 (2%)	2 (17%)	9 (11%)	*0.65*
Others	n (%)	12 (9%)	2 (5%)	3 (25%)	7 (9%)	*0.65*
Follow-Up (month)	Median, (range)	8 (1–41)	18 (2–36)	15 (2–41)	5 (1–33)	*<0.001*

CR = complete response; PR = partial response; MR = mixed response; SD = stable disease; PD = progression disease; ECOG–PS = Eastern Cooperative Oncology Group–performance status; eGFR = estimated glomerular filtration rate; BC = bladder carcinoma; UTUC = upper tract urothelial carcinoma.

## Data Availability

Not applicable.

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
