# Peer review of "Organ-Specific and Mixed Responses to Pembrolizumab in Patients with Unresectable or Metastatic Urothelial Carcinoma: A Multicenter Retrospective Study"

_cancers, 2022, doi:10.3390/cancers14071735_

Round 1

Reviewer 1 Report

It is with great pleasure that I reviewed the article entitled ‘Organ-specific and Mixed Responses to Pembrolizumab in Patients with Unresectable or Metastatic Urothelial Carcinoma: A Multicenter Retrospective Study’ by Dr. Shimizu et al. This is a fascinating well written study from a solid group of clinicians out of Nara, Japan. They set out to gain knowledge on those patients who demonstrated a mixed response to ICI. They defined ‘mixed response’ as the finding of simultaneous regressing and progressing metastatic lesions (e.g., PD in one lesion plus either CR or PR in another lesion). This is a common occurrence in clinical practice. 

First the noted something interesting in their cohort; the overall survival (OS) between responders and non-responders with locally advanced lesions and 54 lymph node, lung, or liver metastases (HR 9.02 [3.63–22.4] P < 0.0001; HR 3.63 [1.97–6.69], P < 0.0001; 55 HR 2.75 [1.35–5.59], P = 0.0053; HR 3.17 [1.00–10.0], P = 0.049, respectively). This would suggest to me that before trying ICI, one should consider debulking the primary/locally advanced tumor, if it is still in place.  Very interesting. But let’s turn to their main conclusion; MR was significantly associated with a poorer prognosis than that of the responder group (CR or 58 PR; HR 0.09 [0.02–0.35], P = 0.004). Patients with bone metastases had less benefit. I guess this is largely what would be expected, but it’s good to have higher level evidence now.  This shows then the limitation of RECIST v1.1. So how can we improve? What are your next steps?

TITLE: No issues

ABSTRACT: No issues

INTRO: No issues
MATERIALS AND METHODS: Nicely laid out and simple methods tying into their prospective study.

Good that only pembro was utilized making this a clearer cohort. 

RESULTS: No issues. Well laid out and simple to follow/understand. 

DISCUSSION: What is MRI in the second paragraph? Perhaps this section needs editing. See the sentence near the end of the second paragraph…’Our group has reported that responsiveness to good chemotherapy may 289 also correlate with subsequent responsiveness to pembrolizumab’. What is good chemotherapy? Has anyone looked at the outcomes when these MR are continued with pembro or other ICI vs. go for early next line therapy? Also the conclusion states MRI is common.  I should be MR is common.

REFERENCES: No issues

TABLES:

Table 1 No issues appears to be well balanced

FIGURES:

Figure 1 no issues

Figure 2 no issues

Figure 3 no issues, nice lay out

Figure 4 and 5

Figure 6 ok but #s are very small.

Reviewer 2 Report

  1. Were responses evaluated by central radiology review.
  2. Among MR patients, how many would be SD by routine criteria, and how many PR and PD? Does MR impact outcomes independent of the routine definition?
  3. Did impact of MR differ based on the organ where progression was seen?
  4. Authors could discuss if the MR group reflects pseudoprogression?
  5. The definition of MR is too rigid. Authors could broaden the definition and include those with discordant responses in different organs-especially those with CR/PR and SD- ie CR and SD, PR and SD.
